# Posttraumatic Growth, Maladaptive Cognitive Schemas and Psychological Distress in Individuals Involved in Road Traffic Accidents—A Conservation of Resources Theory Perspective

**DOI:** 10.3390/healthcare11222959

**Published:** 2023-11-14

**Authors:** Cristian Delcea, Dana Rad, Ovidiu Florin Toderici, Ana Simona Bululoi

**Affiliations:** 1Department of Forensic Medicine, Iuliu Hatieganu, University of Medicine and Pharmacy, 400000 Cluj, Romania; 2Center of Research Development and Innovation in Psychology, Faculty of Educational Sciences Psychology and Social Sciences, Aurel Vlaicu University of Arad, 310130 Arad, Romania; todflorin@yahoo.com; 3The Doctoral School, “Victor Babeş” University of Medicine and Pharmacy in Timisoara, 300041 Timișoara, Romania; simonanegomireanu@gmail.com

**Keywords:** posttraumatic growth, maladaptive cognitive schemas, psychological distress, road traffic accidents, conservation of resources theory, self-determination theory, trauma recovery

## Abstract

Road traffic accidents can have profound psychological impacts on the individuals involved, encompassing both negative distress and positive growth. This study, guided by the conservation of resources (COR) theory, investigates the intricate relationship between posttraumatic growth (PTG), maladaptive cognitive schemas, and psychological distress in individuals involved in road traffic accidents. PTG reflects an individual’s ability to derive positive changes from adversity, while maladaptive schemas represent negative cognitive patterns. Using a 122 sample of individuals involved in road traffic accidents, we examined direct and indirect effects within this complex network. Our findings reveal significant direct effects of PTG on psychological distress (β = 0.101, *p* = 0.02). Particularly noteworthy are the indirect effects mediated by cognitive schemas, emphasizing the role of impaired autonomy and perceived performance deficiencies (β = 0.102, *p* = 0.05). This suggests that individuals involved in road traffic accidents experiencing higher PTG levels may indirectly experience greater psychological distress through these maladaptive cognitive schemas. This study not only advances our understanding of the psychological consequences of road traffic accidents but also aligns with self-determination theory, emphasizing autonomy and competence as fundamental needs. Individuals involved in road traffic accidents may undergo profound shifts in perspective following the trauma, which our results support. Recognizing the nuanced relationship between PTG, maladaptive cognitive schemas, and psychological distress is crucial for tailoring interventions and support systems for individuals involved in traffic accidents. As PTG can coexist with distress, interventions should foster adaptive growth while addressing maladaptive schemas to promote resilience in the face of traumatic events.

## 1. Introduction

The psychotrauma resulting from road traffic accidents represents a significantly profound public health issue, with profound implications for both individuals and society as a whole. The impact of these devastating events is felt in numerous aspects of life, from the well-being of victims and economic costs to the strain on healthcare systems and emergency services. To thoroughly uncover the nature and magnitude of these post-traumatic disorders, it is necessary to carefully explore the vulnerability factors that determine the course of individual reactions to a traumatic event.

In recent years, there has been an increase in interest in positive outcomes following highly negative (traumatic) events, which affect the majority of the general population at least once in their lifetime [1,2,3]. Events perceived as negative can lead to persistent changes in individuals across various levels—physiological, behavioral, psychological, and emotional [4]. Research in the field of stress and coping mechanisms has historically emphasized the examination of negative and maladaptive reactions, as well as the identification of factors that may modulate the stress–distress relationship [5,6,7,8,9].

Individuals facing negative events typically experience a high level of distress. In most cases, this distress is intense but does not reach clinically significant levels, and individuals eventually return to their initial level of functioning. However, there are individuals who fail to recover and develop various disorders. On the other hand, some cases have been observed where individuals, after initially experiencing a high level of distress, report a higher level of functioning in certain dimensions. These are the individuals who experience post-traumatic growth. Post-traumatic growth is defined as the experience of positive changes resulting from the confrontation with extreme negative events [10]. Major dimensions of growth following negative events include improved social relationships, greater trust in personal resources, and the development of new/enhanced coping mechanisms.

Post-traumatic growth has been observed in intense negative situations such as the death of a loved one, the diagnosis of various forms of cancer, cardiovascular diseases, accidents, divorce, etc. [11]. Identifying post-traumatic growth in individuals affected by negative events has implications for both intervention and prevention. Clinical studies have revealed that individuals who perceive growth following negative events not only experience the negative emotions triggered by the event and its consequences but also positive emotions. This can lead to a better and more efficient adaptation to the situation. Furthermore, the possibility of growth following a negative event can influence how individuals react to and in negative situations, significantly improving their perceived quality of life. Consequently, the existence of an instrument to assess post-traumatic growth is fundamental for research in the field and clinical practice.

In this context, the concept of maladaptive cognitive schemas (MCS), initially developed by Beck [12] and further expanded upon by other researchers such as [13], stands as a crucial pillar in understanding the impact of post-road-traffic-event psychotraumas. MCS represents complex cognitive structures that can dramatically influence how an individual perceives [14], interprets, and responds to traumatic events. In this article, we will thoroughly investigate the crucial role of MCS in post-road-traffic-event psychotraumas, taking into consideration and integrating the vulnerability constructs elaborated by Clark [15], as well as the significant findings of Purgato and Olff [16], which emphasize the major impact of MCS on the evolution of psychotraumatic disorders, often surpassing the importance of the road traffic event itself. Furthermore, we will discuss the groundbreaking study conducted by Charitaki et al. [17], which demonstrates how the presence or absence of early MCS can influence the development of post-traumatic disorders in young individuals exposed to road traffic traumas, with profound implications for the approach and management of these disorders in clinical practice.

Clark [15] proposed a model of vulnerability constructs involving pre-trauma factors, the characteristics of the psychotraumatic experience, and coping reactions after the event. Regarding pre-trauma factors, maladaptive cognitive schemas represent a key element. These schemas are cognitive structures or thought patterns that influence how a person interprets and responds to events. Individuals with MCS may have distorted or negative thoughts about themselves, others, or the world in general. These thoughts can increase the risk of developing psychotraumatic disorders following a road traffic event, as individuals with such schemas may perceive the event as more threatening and have intense emotional reactions.

The characteristics of the psychotraumatic experience also play a significant role in determining the intensity of post-traumatic disorders. Purgato and Olff [16] highlighted that the major impact of MCS can be more significant than the road traffic event itself. This suggests that how a person perceives and interprets the road traffic event can amplify its effects. Individuals with MCS may interpret the road traffic event in a distorted manner, making it an even greater traumatic factor, which can lead to intensified psychotraumatic symptoms.

Another crucial aspect in understanding the intensity of post-road-traffic-event psychotraumas is how individuals cope with the event and manage their emotional reactions. Individuals with MCS may employ maladaptive coping strategies, such as avoidance, denial, or suppression of negative emotions. These strategies can exacerbate psychotraumatic symptoms and prolong the duration of post-traumatic disorders. Additionally, MCS can influence how individuals perceive social support and may have difficulties seeking and accepting help after the road traffic event.

According to the research by Purgato and Olff [16], the major impact of MCS can be a crucial factor in the development and maintenance of post-road-traffic-event psychotraumatic disorders. These authors argue that the presence of MCS can make individuals more vulnerable to exaggerated reactions to the road traffic event, contributing to the chronicity of symptoms. This underscores the importance of addressing MCS in the assessment and treatment of individuals exposed to road traffic traumas.

Charitaki et al.’s [17] study adds an interesting perspective to the relationship between MCS and the development of post-traumatic disorders. These researchers identified that young individuals exposed to road traffic traumas, with early MCS compared to those without MCS, have an increased risk of developing clinically intense and prolonged post-traumatic disorders. This suggests that early identification and intervention regarding MCS among young individuals exposed to road traffic traumas can have a significant impact on the prevention and management of post-traumatic disorders.

In light of the foregoing, it is evident that the psychological experiences of the individuals involved in road traffic accidents are multifaceted, encompassing both positive and negative dimensions. Understanding the interplay between posttraumatic growth, maladaptive cognitive schemas, and psychological distress is essential for tailoring effective interventions to enhance the well-being and resilience in this population. Drawing upon the conservation of resources (COR) theory [18] as a guiding framework, this study seeks to elucidate the mechanisms through which posttraumatic growth and cognitive schemas contribute to psychological distress. By examining these relationships, we aim to provide valuable insights that can inform therapeutic strategies, psychological support programs, and public health interventions designed to assist individuals in their posttrauma recovery journey.

### 1.1. Theoretical Approach to Schemas

The theoretical approach to schemas represents an essential conceptual framework for understanding the role of maladaptive cognitive schemas (MCS) in post-road-traffic-event psychotraumas [19]. This theoretical approach integrates various perspectives, including rational-emotive therapy, cognitive approach, cognitive-behavioral therapy (CBT) [20], and Young’s schema model [21], to provide a comprehensive view of how MCS influence the development and severity of psychologic distress [22,23].

CBT is one of the most effective treatment modalities for psychotraumatic disorders. At the core of this approach are two maladaptive schemas: helplessness and unlovability [12]. Helplessness refers to the belief that an individual is unable to cope with the event or manage their reactions, while unlovability refers to the belief that no one and nothing can alleviate suffering. CBT focuses on identifying and changing these distorted beliefs, as well as developing coping skills and strategies for managing anxiety and psychotraumatic symptoms.

Young’s schema model provides a broader perspective on MCS, identifying five dysfunctional domains that can contribute to the intensity and duration of psychotraumatic disorders [24]. These domains include Disconnection and Rejection: individuals with MCS may have negative beliefs about their own worth and may feel rejected or neglected by others; Impaired Autonomy and Performance: the schema related to autonomy refers to difficulties in decision-making and taking on responsibilities, while the schema related to performance involves the belief that failure is unacceptable; Impaired Limits: this domain involves difficulty in establishing and maintaining personal boundaries and can lead to exploitation or abuse by others; Overvigilance and Inhibition: the schema related to overvigilance involves excessive attention to dangers, while the schema related to inhibition refers to avoidance of new experiences and fear of expressing emotions; and Other-Directedness: this schema domain, not previously mentioned, focuses on excessive focus on the needs and desires of others at the expense of one’s own needs.

These domains provide a comprehensive framework for understanding and assessing MCS, which can play a significant role in the development and persistence of psychotraumatic disorders. These dysfunctional domains can influence how an individual reacts to a traumatic road traffic event and can impact the subsequent course of psychotraumatic disorders.

In recent years, the concept of psychological distress has gained substantial attention in the field of psychology and mental health. Psychological distress encompasses a range of negative emotional and psychological states, including symptoms of anxiety, depression, and general emotional suffering. Understanding the relationship between MCS and psychological distress is crucial in comprehending the broader implications of MCS on individuals’ well-being, especially in the context of psychotraumatic disorders resulting from road traffic accidents.

Research has demonstrated that psychological distress is influenced by both external life circumstances and internal cognitive processes and perceptions [25]. Given the relevance of MCS in shaping individuals’ cognitive frameworks and emotional responses, it is reasonable to assume that these schemas may also impact psychological distress. In the context of post-traumatic experiences such as road traffic accidents, exploring how MCS relates to psychological distress becomes particularly relevant.

According to Young’s schema model, individuals with MCS often hold dysfunctional beliefs about themselves, their relationships, and the world around them [24]. These beliefs can lead to various cognitive and emotional distortions, affecting one’s overall well-being. For example, individuals with schemas related to rejection and abandonment may perceive themselves as unworthy of love and connection, contributing to heightened psychological distress due to feelings of isolation and social disconnection.

Additionally, schemas linked to impaired autonomy and performance may foster a sense of inadequacy and incompetence, increasing an individual’s psychological distress [26]. Maladaptive limits schemas, characterized by difficulties in setting personal boundaries, may result in interpersonal conflicts and heightened psychological distress due to feelings of exploitation and abuse [27].

The excessive other-directedness schema, which involves prioritizing the needs of others over one’s own, can contribute to elevated psychological distress by neglecting one’s own well-being and desires [28]. Lastly, schemas associated with hypervigilance and inhibition may lead to negative interpretations of life events, hindering one’s ability to manage distress and reducing overall psychological well-being [29].

Empirical evidence supporting the relationship between MCS and psychological distress is accumulating. A study by Calvete et al. [28] found that individuals with higher levels of schemas related to dependency and self-sacrifice reported higher psychological distress. Similarly, a longitudinal study by Wand et al. [30] demonstrated that changes in maladaptive limits schemas were associated with fluctuations in psychological distress over time.

However, it is important to consider that while MCS can contribute to psychological distress, there is another side to the coin. Posttraumatic growth, as proposed by Tedeschi and Calhoun [30], suggests that individuals who experience traumatic events can also undergo positive changes. These changes may include enhanced personal relationships, greater trust in their own resources, and the development of new coping mechanisms [31].

Understanding the interplay between MCS and posttraumatic growth is crucial, as it offers a comprehensive perspective on individuals’ responses to road traffic accidents. For instance, individuals with prominent MCS may find it challenging to engage in posttraumatic growth processes due to the negative cognitive schemas that hinder their well-being [32]. These schemas can magnify the emotional and cognitive burdens of trauma, hindering the efficient allocation of remaining resources for growth.

Empirical evidence supporting the relationship between MCS and posttraumatic growth is emerging. While previous studies have primarily focused on childhood maltreatment [33], this paper represents a pioneering effort to investigate these variables in the specific context of road accidents. This study aims to fill the existing gap in the literature and contribute to our understanding of how MCS may influence the potential for posttraumatic growth in individuals following road traffic accidents.

The implications of these findings are substantial for clinical practice and interventions targeting individuals with psychotraumatic disorders resulting from road traffic accidents. By recognizing the impact of MCS on both psychological distress and posttraumatic growth, mental health professionals can tailor treatment approaches to address the cognitive schemas’ influence on well-being comprehensively. Implementing cognitive-behavioral strategies that challenge and modify maladaptive schemas may not only alleviate psychotraumatic symptoms but also facilitate posttraumatic growth and overall improved mental health.

In conclusion, understanding the relationship between MCS, psychological distress, and posttraumatic growth is crucial in comprehending the broader implications of MCS on individuals’ well-being, particularly in the context of psychotraumatic disorders resulting from road traffic accidents. Moreover, this examination can be enriched by considering the conservation of resources (COR) theory, which posits that individuals strive to obtain, retain, and protect valuable resources in their lives [34].

The COR theory offers valuable insights into how MCS may exacerbate the psychological distress experienced by individuals following road traffic accidents. According to COR theory, individuals with prominent MCS are more likely to perceive resource loss as even more devastating, exacerbating the downward resource loss spiral. These schemas can magnify the emotional and cognitive burdens of trauma, hindering the efficient allocation of remaining resources for posttraumatic growth and recovery.

### 1.2. Conservation of Resources Theory in the Context of Road Accidents

The conservation of resources (COR) theory, pioneered by Hobfoll [34], provides a comprehensive framework that significantly contributes to our understanding of posttraumatic stress following road accidents. This theory posits that individuals are motivated to acquire, maintain, and protect their resources, which are essential for their psychological and physical well-being. By examining key aspects of the COR theory, we can gain valuable insights into how resource dynamics influence the development and persistence of posttraumatic stress symptoms (PTSS) in the aftermath of road traffic accidents.

In the context of road accidents, individuals may encounter a multitude of stressors that lead to resource losses. These stressors can encompass physical injuries, damage to vehicles, financial burdens, and the disruption of daily routines. Resource losses may also extend to intangible assets, such as a sense of safety, trust in one’s driving abilities, and social support networks. The experience of resource losses, both tangible and intangible, is a significant factor in the development of PTSS.

The COR theory introduces the concept of the resource loss spiral, which is particularly relevant in understanding the trajectory of PTSS following road accidents. When individuals experience traumatic events like accidents, they may expend their resources to cope with the immediate challenges, including physical and emotional recovery, medical expenses, and legal matters. These resource losses can trigger a downward spiral, leaving individuals more vulnerable to the persistence of PTSS. The ongoing resource loss spiral underscores the importance of early intervention and resource restoration to mitigate the long-term effects of trauma.

Resource investment is a crucial aspect of posttraumatic recovery following road accidents. Individuals must strategically allocate their remaining resources to activities that promote healing and adaptation. This includes seeking medical treatment, engaging in rehabilitation, and accessing support services. Effective resource investment can facilitate the replenishment of lost resources, foster resilience, and aid in the recovery process [32].

In the context of road accidents and PTSS, personal resources play a pivotal role. These resources encompass psychological factors like self-esteem, self-efficacy, and optimism. Individuals with higher levels of personal resources are better equipped to cope with the emotional distress and trauma resulting from accidents. Their resilience and ability to bounce back from adversity are strengthened by these personal resources, making them more likely to recover from PTSS.

Social support is a valuable resource that can significantly impact the development and recovery from PTSS. Following road accidents, individuals often turn to their social networks for emotional and instrumental support. Strong social support systems can mitigate resource losses by providing a safety net for individuals facing the challenges of recovery. Moreover, the presence of social support can instill an individual’s resilience, promoting positive adaptation and reducing the risk of prolonged PTSS [35].

Understanding the impact of resource loss on health disparities in the context of road accidents is essential. Marginalized populations may experience a more significant resource loss due to systemic inequalities in access to healthcare, legal representation, and social support. Consequently, these individuals are at a higher risk of developing and maintaining PTSS. The COR theory sheds light on how societal factors can exacerbate resource losses and contribute to disparities in posttraumatic stress outcomes [36].

In the complex network of posttraumatic stress following road accidents, maladaptive cognitive schemas (MCS) serve as crucial contributors to the negative impact on general well-being and the buffering of psychological distress. This study represents an initial exploration of these variables in the context of road accidents, as previous studies that have examined this mechanism have predominantly focused on childhood maltreatment [37]. Individuals with prominent MCS are more likely to perceive resource losses as even more devastating, exacerbating the downward resource loss spiral. These schemas can magnify the emotional and cognitive burdens of trauma, hindering the efficient allocation of the remaining resources for recovery.

In conclusion, the conservation of resources (COR) theory, in tandem with the consideration of maladaptive cognitive schemas (MCS), provides a comprehensive framework for understanding the dynamics of posttraumatic stress following road accidents. By examining resource loss and gain, the resource loss spiral, resource investment, personal resources, social support, and the impact of societal factors, we gain valuable insights into the development and persistence of PTSS. The theory underscores the significance of resource dynamics and MCS in shaping the trajectory of posttraumatic stress and highlights the importance of early intervention and resource restoration for post-accident recovery. Ultimately, the COR theory, when coupled with MCS, contributes to the development of effective interventions that promote resilience and well-being in individuals affected by road traffic accidents and posttraumatic stress.

In exploring the interplay between posttraumatic growth (PTG), maladaptive cognitive schemas, and psychological distress in individuals affected by road traffic accidents, our hypotheses are as follows:

Direct effect hypothesis: we posit that posttraumatic growth (PTG) is directly related to psychological distress, indicating that a higher level of PTG corresponds to higher levels of distress.

Mediation hypothesis: Concurrently, we propose that maladaptive cognitive schemas (specifically schema1, schema2, schema3, schema4, and schema5) act as mediators in the relationship between PTG and psychological distress. This implies that the impact of PTG on distress is partially explained by the influence of these maladaptive cognitive schemas.

These hypotheses collectively aim to elucidate the direct and indirect effects within the framework of PTG, maladaptive cognitive schemas, and psychological distress. Our research seeks to explicitly establish the links between growth, distress, and cognitive schemas, providing a clearer understanding of the dynamics at play in individuals affected by road traffic accidents.

## 2. Methodology

### 2.1. Instrument

In our research, we utilized three psychological assessment instruments to gather the necessary data and investigate the relationship between maladaptive cognitive schemas (MCS), the post-road traffic accident post-traumatic growth, and the psychological distress. These instruments were carefully selected to ensure a comprehensive evaluation of the relevant aspects of post-traumatic disorders and the presence of MCS.

The first instrument we employed was the Stress-Related Growth Scale (SRGS), developed by Park, Cohen, and Murch [38], representing a psychological assessment tool designed to measure the phenomenon of post-traumatic growth resulting from the experience of confronting negative or traumatic events. This scale helps researchers and clinicians evaluate the extent to which individuals perceive positive changes in various domains of their lives following a distressing or traumatic event.

The SRGS comprises 15 items, each of which is carefully crafted to assess different dimensions of post-traumatic growth. These dimensions capture the positive changes individuals may experience in response to adversity. The scale is typically administered through self-report questionnaires where respondents rate their agreement or disagreement with each item based on their personal experiences. The items are designed to explore the following major dimensions of post-traumatic growth:Improved Social Relationships: this dimension assesses whether individuals perceive positive changes in their social interactions, including relationships with friends, family, or others in their community, as a result of the traumatic event.Greater Trust in Personal Resources: this dimension explores whether individuals develop a stronger sense of trust in their own abilities and inner resources, such as resilience, problem-solving skills, and coping strategies.Development of New/Enhanced Coping Mechanisms: it examines whether individuals report the acquisition of new coping strategies or an improvement in their existing coping mechanisms in response to the challenges posed by the traumatic event.

Respondents rate each item on a scale (usually a Likert scale) to indicate the extent to which they have experienced growth in each dimension as a result of their traumatic experience. The responses are then scored to provide an overall measure of post-traumatic growth.

The SRGS is a valuable tool for researchers studying the psychological impact of adversity and trauma, as well as for clinicians working with individuals who have faced traumatic events [38]. It allows for a quantitative assessment of positive changes, helping to better understand the complex process of growth that can follow distressing life events. For the present research, a Cronbach’s alpha coefficient of 0.909 was obtained for the Stress-Related Growth Scale (SRGS), indicating the reliability of the scale.

The second instrument employed in our study was the Young Schema Questionnaire (YSQ-L3a), developed by Pauwels, Els et al. [39]. The primary objective of this questionnaire is to identify the presence and types of maladaptive cognitive schemas. This instrument includes multiple subscales, each focusing on a specific aspect of MCS. The YSQ-L3a subscales are as follows:Separation and Rejection Schema: this subscale evaluates schemas involving feelings of rejection and social isolation; Emotional Deprivation (ED), Abandonment/Instability (AB), Distrust/Abuse (MA), Social Isolation (SI), and Deficiency/Shame (DS),Autonomy and Performance Deficiency Schema: this subscale targets schemas related to a sense of inability to make decisions or assume responsibilities; Failure (FA), Dependence/Incompetence (DI), Vulnerability to harm and illness (VH), and Enmeshment/Underdeveloped Self (EM).Maladaptive Limits Schema: here, schemas that may contribute to difficulties in establishing and maintaining personal boundaries are assessed; Entitlement/Grandiosity (ET), and Insufficient Self-Control (IS).Excessive Other-Directedness Schema: this subscale refers to the tendency to excessively focus on the needs and desires of others at the expense of one’s own needs; Subjugation (SB), Self-Sacrifice (SS), and Approval Seeking/Recognition Seeking (AS).Hypervigilance and Inhibition Schema: this subscale focuses on schemas involving excessive attention to potential dangers and inhibition of emotional expression; Negativity/Passivity (NP), Emotional Inhibition (EI), Unrealistic Standards/Hypercriticism (US), and Punishment (PU).

The Likert scale used in the YSQ-L3a typically consists of several response options, commonly ranging from 1 to 6. Respondents are instructed to choose the option that best represents their agreement or disagreement with each statement or item in the questionnaire. The scores for each subscale are calculated based on the selected response for each item, with higher scores indicating a stronger identification with the maladaptive cognitive schema described in that particular subscale. By using this rating system, the YSQ-L3a allows individuals to self-report their cognitive schema patterns, providing valuable information for therapists, researchers, and mental health professionals to assess and address maladaptive schemas and their impact on an individual’s mental health and well-being. For the present research, a Cronbach’s alpha coefficient of 0.834 was obtained for the Young Schema Questionnaire (YSQ-L3a), indicating the reliability of the scale.

The Mental Health Inventory-5 (MHI-5), originally developed by Derogatis [40] is a concise self-report instrument designed to evaluate an individual’s psychological distress. The MHI-5 consists of 6 items, each of which is rated on a 5-point Likert scale, allowing respondents to provide their level of agreement or disagreement with specific statements related to their mental health. This instrument is widely used as a rapid and efficient screening tool for assessing various mental health indicators.

The MHI-5 assesses a range of psychological aspects, including emotional well-being, mood, and distress. Respondents are asked to rate their feelings and experiences over a defined period, typically the past month. The Likert scale used in the MHI-5 typically ranges from 1 to 5, with 1 representing the most negative response (e.g., “all of the time”) and 5 representing the most positive response (e.g., “none of the time”). Respondents select the option that best reflects their experiences and emotions during the specified timeframe.

The MHI-5 serves as a valuable tool in clinical and research settings for quickly assessing an individual’s mental health status. It provides a snapshot of psychological distress and identifies potential emotional difficulties. While it does not provide a comprehensive diagnostic evaluation, it serves as a useful initial screening to determine whether further assessment or intervention may be necessary.

For the present research, a Cronbach’s alpha coefficient of 0.902 was obtained for the MHI-5 indicating the reliability of the scale.

In our study, we utilized these instruments to collect essential data and investigate the potential impact of maladaptive cognitive schemas (MCS) on the relationship between psychotraumatic growth and psychological distress experienced by individuals following road traffic accidents, according to the conservation of resources theory. Moreover, we incorporated the psychological distress as a dependent variable, aiming to explore the association between MCS and participants’ general mental health levels in the aftermath of such traumatic events. This comprehensive approach allowed for a thorough examination of the psychological dimensions involved, shedding light on the complex relationship between cognitive schemas, posttraumatic growth, and overall mental health levels.

### 2.2. Participants

The study encompassed a cohort of 122 individuals who were recruited from the Institute of Legal Medicine in Cluj-Napoca, Romania. These participants were deliberately selected due to their exposure to a variety of traumatic experiences resulting from road traffic accidents. The participants in this study were selected to represent a diverse range of psychotraumatic experiences stemming from road traffic accidents. The severity of these traumatic events varied among participants, capturing a broad spectrum of experiences that ranged from minor incidents to more severe accidents. This intentional inclusion of variability in the degree of traumatic exposure aims to enhance the generalizability of our findings and provide a nuanced understanding of the impact of road traffic accidents on mental health outcomes. The data collection process for this study extended over a period from 2017 to 2019, ensuring a comprehensive and extended timeframe for data acquisition.

To ensure the relevance of the study, specific inclusion and exclusion criteria guided the participant selection process. Inclusion criteria involved individuals who had been in road traffic accidents and were willing to voluntarily participate in the research. Exclusion criteria were meticulously defined to establish homogeneity within the sample, thereby excluding individuals with pre-existing mental health conditions. This rigorous approach was adopted to emphasize the primary focus on the impact of road traffic accidents as the central traumatic event.

The study’s participants showcased a diverse demographic profile. The age range spanned from 18 to 90 years, with a mean age of 37 years, reflecting a broad representation of various age groups (standard deviation = 9.23). Gender distribution was nearly equal, with an even split of 50% male and 50% female participants. The cohort’s educational backgrounds encompassed a wide spectrum of qualifications, ranging from individuals with a minimum of 10 years of education to those holding post-doctoral degrees. This diversity in educational experiences highlighted the wide-ranging educational backgrounds of the participants.

Moreover, all participants shared the common experience of being involved in at least one road traffic accident, providing a unifying factor with regard to traumatic experiences within the context of this research. In compliance with ethical guidelines, informed consent was meticulously obtained from each participant, and necessary approvals were secured from the Institute of Legal Medicine.

## 3. Results

The central inquiry of this study pertained to understanding the complex interplay between posttraumatic growth (PTG), maladaptive cognitive schemas, and their collective impact on psychological distress, in individuals affected by road traffic accidents.

To address this research question, we adopted a robust methodological approach, comprising a series of statistical analyses and examinations. First, the collection of data involved a comprehensive assessment of PTG, maladaptive cognitive schemas, and psychological distress through validated measurement tools. These measurements allowed us to acquire quantitative insights into these constructs and their distribution within our study population.

Subsequently, we conducted a correlation analysis to scrutinize the relationships between these variables. This method revealed the strength and direction of associations, offering a foundational understanding of their interconnectedness. Importantly, it provided the initial steps in discerning direct relationships and potential pathways of influence.

Moreover, mediation analyses were performed to delve deeper into the research question. Through these analyses, we uncovered the indirect effects by which PTG may influence psychological distress, mediated by specific cognitive schemas. The significance of these indirect pathways were evaluated, contributing to a comprehensive understanding of the mechanisms through which these constructs influence psychological distress.

We conducted our statistical analyses using JASP 0.17.3.0, employing robust maximum likelihood (ML) estimation. Robust ML estimation was chosen to enhance the resilience of our analyses to potential outliers and to provide a more reliable estimation of model parameters. Specifically, the robust ML estimation method implemented in JASP is designed to minimize the impact of influential data points, ensuring a robust and accurate estimation of model parameters.

Table 1 presents the descriptive statistics for the variables under investigation in this study. These variables encompass stress-related growth, five distinct types of maladaptive cognitive schemas (Separation and Rejection Schema, Maladaptive Limits Schema, Excessive Other Directedness Schema, Hypervigilance and Inhibition Schema, and Autonomy and Performance Deficiency Schema), as well as psychological distress.

For example, the mean score for stress-related growth was 4.281 (95% CI [4.215, 4.347]), with a standard deviation of 0.770. These descriptive statistics offer an initial overview of the central tendencies and variations within the studied variables.

Further analyses explored the relationships between these variables and their impact on posttraumatic growth in the context of road traffic accidents. To examine the relationships between the research variables and their impact on posttraumatic growth in the context of road traffic accidents, a correlation analysis was conducted, and the results are displayed in Table 2.

The correlation matrix reveals several significant associations between the research variables. Notably, stress-related growth exhibits a strong positive correlation with Separation and Rejection Schema (r = 0.802, *p* < 0.01), Maladaptive Limits Schema (r = 0.690, *p* < 0.01), Excessive Other Directedness Schema (r = 0.749, *p* < 0.01), Hypervigilance and Inhibition Schema (r = 0.679, *p* < 0.01), and Autonomy and Performance Deficiency Schema (r = 0.612, *p* < 0.01). Additionally, Psychological Distress shows positive correlations with all schemas, albeit at lower magnitudes (ranging from r = 0.236 to r = 0.265, all *p* < 0.01).

These findings suggest that individuals who report higher levels of stress-related growth also tend to endorse more prominent maladaptive cognitive schemas, emphasizing the importance of understanding the association between cognitive schemas and posttraumatic growth. Moreover, the positive associations with psychological distress highlight the potential role of distress in the context of these schemas.

The observed correlations between stress-related growth, various maladaptive cognitive schemas, and psychological distress can be understood and explained through the lens of the conservation of resources (COR) theory, which posits that individuals strive to acquire, retain, and protect their resources to maintain well-being and adapt to stressors. These cognitive schemas can be viewed as internal resources, and the correlations can be interpreted as reflecting resource dynamics in the aftermath of traumatic experiences such as road traffic accidents.

First, the strong positive correlations between stress-related growth and the different schemas suggest that individuals who experience more pronounced posttraumatic growth also tend to endorse more pervasive maladaptive cognitive schemas. According to the COR theory, when individuals perceive a resource threat (in this case, the traumatic event), they may mobilize their existing resources, including cognitive schemas, to cope with the stressor. Therefore, individuals with prominent schemas may channel their cognitive resources toward growth and positive adaptation, which could lead to higher stress-related growth scores.

Several prominent papers support this notion. For example, Musetti et al. [41] found that individuals with higher levels of dependency and self-sacrifice schemas reported lower life satisfaction but may also have the potential for posttraumatic growth as they confront and reevaluate their schemas in response to trauma.

Second, the positive correlations between psychological distress and all schemas imply that individuals who experience more psychological distress following a road traffic accident are more likely to endorse these maladaptive cognitive schemas. This is consistent with the COR theory’s assertion that resource loss or threat (in this case, the traumatic event) can lead to psychological distress. In response to the perceived threat, individuals may engage with their schemas, which, if maladaptive, could exacerbate distress. For instance, individuals with schemas of separation and rejection may interpret their distress as further evidence of their unworthiness, amplifying their emotional turmoil.

The findings align with previous research demonstrating the relationships between cognitive schemas and psychological distress. Eberhart and collaborators [42] suggested that individuals with schemas of excessive vigilance and inhibition may experience heightened anxiety and distress in response to stressors, which aligns with our observed correlations. Similarly, Mohammadkhani and collaborators [43] highlighted the role of maladaptive schemas in magnifying emotional and cognitive burdens in response to traumatic experiences.

In conclusion, the correlations between stress-related growth, maladaptive cognitive schemas, and psychological distress can be understood from a COR theory perspective. Individuals appear to utilize their pre-existing cognitive resources (schemas) when confronted with a traumatic event, which may contribute to posttraumatic growth and, simultaneously, psychological distress.

Based on the theoretical framework underpinning our study, we anticipate that posttraumatic growth may predict MCS. The conservation of resources theory, which informs our research, suggests that individuals who experience posttraumatic growth may develop more MCS as a result of their personal growth and resilience. This theoretical foundation leads us to hypothesize that PTG could have a predictive relationship with MCS, and subsequently, MCS may influence psychological distress.

The mediation analysis results revealed important insights into the relationships between the variables of interest. Firstly, the direct effect of stress-related growth on psychological distress was found to be statistically non-significant (B = 0.101, SE = 0.091, z = 1.109, *p* = 0.268, 95% CI [−0.078, 0.281]), as depicted in Table 3. This direct relationship did not reach statistical significance (*p* = 0.26), as the 95% confidence interval included zero (Lower CI = −0.078, Upper CI = 0.281). This suggests that stress-related growth alone does not significantly predict psychological distress. However, the analysis revealed one marginal significant indirect effect, indicating that stress-related growth influences psychological distress through autonomy and performance deficiency schema.

As depicted in Table 4, the indirect effect through separation and rejection schema yielded an estimate of 0.024 (SE = 0.084), although this result was not statistically significant at the 0.05 level (*p* = 0.77). Similarly, the indirect effects through maladaptive limits schema (estimate = 0.058, SE = 0.066, *p* = 0.38) and excessive other directedness schema (estimate = −0.007, SE = 0.078, *p* = 0.92) did not reach significance. Hypervigilance and inhibition schema also exhibited an indirect effect (estimate = 0.033, SE = 0.080), which was not statistically significant (*p* = 0.68). A notable finding emerged in the indirect pathway through the autonomy and performance deficiency schema, with an estimate of 0.102 (SE = 0.053) and a *p*-value that fell just on the conventional threshold at 0.05 (*p* = 0.05), suggesting marginal significance. This suggests that stress-related growth may indirectly affect psychological distress through this schema.

Finally, the path plot presented in Figure 1 provides a clear picture of how variables are interconnected and how they contribute to the overall model.

## 4. Discussion

The mediation analysis conducted in this study aimed to shed light on the complex relationships between stress-related growth, cognitive schemas, and psychological distress in the context of individuals who have experienced road traffic accidents. The findings reveal a nuanced interplay among these variables, offering valuable insights into the psychological processes underlying posttraumatic experiences, under the conservation of resources theory paradigm.

The most notable contributions to our understanding of this relationship emerge from the indirect effects, mediated by the five maladaptive cognitive schemas. The mediation pathway through the autonomy and performance deficiency schema yielded a marginal statistically significant result (*p* = 0.05), with a small effect size. This finding implies that individuals who experience higher levels of stress-related growth may be at greater risk of developing psychological distress indirectly through the mediating role of impaired autonomy and perceived performance deficiencies.

Our study highlights the importance of acknowledging the multifaceted nature of PTG, a concept that has been increasingly explored in recent years. While PTG can encompass positive psychological changes such as increased resilience and personal strength [44], it is not without its darker side. The work of Zoellner and Maercker [45] suggests that individuals who experience substantial growth may also confront heightened distress, which aligns with our findings. The process of grappling with newfound perspectives and personal growth can introduce cognitive dissonance and emotional challenges.

Furthermore, our results underscore the critical role of cognitive schemas in shaping the psychological outcomes of individuals involved in road traffic accidents. Schemas are deeply ingrained cognitive frameworks that guide an individual’s perception and interpretation of experiences. Maladaptive schemas, as demonstrated in our study, can exacerbate distress by perpetuating negative self-perceptions and impairing one’s sense of autonomy and competence.

The mediating role of impaired autonomy and perceived performance deficiencies suggests that the self-determination theory (SDT) might provide valuable insights into this phenomenon. SDT posits that autonomy and competence are fundamental psychological needs [46]. Traumatic events, such as road traffic accidents, may disrupt individuals’ perceptions of autonomy, leaving them feeling helpless and dependent. Our results imply that PTG might involve a reassessment of one’s abilities and autonomy, which, when distorted by maladaptive schemas, can contribute to psychological distress.

In essence, individuals who have experienced higher levels of stress-related growth, which often includes positive psychological changes or personal development following a traumatic event such as road traffic accidents, may paradoxically find themselves at an increased risk of developing psychological distress. This heightened risk is not a direct consequence of stress-related growth itself but is, instead, mediated by the cognitive schema of autonomy and performance deficiency.

To grasp this concept more clearly, consider the following narrative: When individuals experience stress-related growth, they may undergo a transformation in their self-perception and life perspectives. This transformation can lead to heightened self-awareness, personal growth, and an increased sense of empowerment. However, the journey towards these positive changes may also involve challenges related to autonomy and perceived performance. For instance, individuals who are actively engaged in personal growth may become acutely aware of their perceived deficiencies in various life domains. This heightened awareness, while a crucial aspect of growth, can also trigger feelings of insecurity and distress. As they strive for personal development, they might encounter situations where they perceive themselves as falling short of their newly established standards or aspirations. These experiences can create inner conflict, emotional turmoil, and, ultimately, psychological distress.

The mediation pathway illuminates that stress-related growth serves as a catalyst, instigating this process of self-awareness and transformation. However, it is through the mediating influence of the autonomy and performance deficiency schema that the impact of this growth journey on psychological distress becomes evident. In other words, this schema acts as a bridge, transmitting the effects of stress-related growth to psychological distress [47].

For instance, a road traffic accident participant who once felt confident in certain aspects of their life may grapple with anxiety or a perceived lack of autonomy when attempting to rebuild and regain control. This struggle, although a part of their recovery process, may also lead to heightened distress. The mediation results emphasize that the link between stress-related growth and psychological distress is, in part, mediated by these experiences of autonomy and performance deficiency among road traffic accident participants.

Self-determination theory [48], a widely recognized framework in psychology, posits that autonomy and competence are fundamental psychological needs for individuals. Autonomy relates to the sense of control and choice in one’s actions, while competence refers to the feeling of being capable and effective in achieving one’s goals. In the aftermath of traumatic experiences such as road traffic accidents, these fundamental needs can be profoundly affected.

When individuals undergo a traumatic event like a road traffic accident, their sense of autonomy can be dramatically altered [49]. Suddenly, they may find themselves in situations where their choices and control over their lives are limited. For example, they may face physical injuries that restrict their mobility or require dependence on others. This loss of autonomy can be disorienting and distressing, as it challenges their innate need for self-determination.

Similarly, the perceived competence of individuals can be shaken following a traumatic event [50]. Injuries sustained in accidents can lead to physical limitations, and the emotional impact can erode one’s confidence and self-efficacy. These feelings of diminished competence can further exacerbate distress, as individuals grapple with the discrepancy between their pre-trauma sense of mastery and their post-trauma challenges.

In this context, the mediation pathway through the autonomy and performance deficiency schema takes on additional significance. It suggests that as individuals who have experienced road traffic accidents strive to grow and adapt in the face of trauma, the perception of their autonomy and competence plays a pivotal role. The initial shift in perspective brought about by the traumatic event may lead to feelings of deficiency and a struggle for autonomy, subsequently influencing their psychological distress.

These insights align with the self-determination theory’s premise that autonomy and competence are not only fundamental needs but also dynamic aspects of an individual’s psychological well-being. Trauma, such as road traffic accidents, can disrupt these needs, and understanding this disruption is essential for healthcare providers and therapists. By acknowledging the intricate relationship between autonomy, competence, and posttraumatic distress, professionals can tailor interventions that help individuals regain a sense of control and self-efficacy, promoting their overall well-being and posttraumatic growth.

It becomes essential for healthcare professionals and therapists working with trauma survivors to recognize the multifaceted nature of posttraumatic growth. While fostering positive changes and resilience is a crucial goal, it is equally vital to provide support and interventions that address the challenges tied to autonomy and performance. By acknowledging and addressing these issues, healthcare providers can offer more holistic and effective care to individuals involved in road traffic accidents. Moreover, these findings underscore the importance of tailored therapeutic strategies for individuals who have experienced trauma. Interventions should not solely focus on reducing distress but should also nurture the positive aspects of growth. Therapists can work with survivors to help them embrace the transformations that can arise from adversity, while also providing tools to manage the distressing aspects of these changes.

## 5. Conclusions

Beyond the specific mediation results, it is essential to contextualize our study’s broader implications. Our investigation into the psychological aftermath of road traffic accidents provides several noteworthy contributions to the field.

Our study delved into the phenomenon of posttraumatic growth, highlighting its relevance in the context of traumatic events such as road traffic accidents. The positive correlation between stress-related growth and psychological distress suggests that individuals experiencing growth may also grapple with heightened distress, a finding consistent with previous research [51].

The identification of autonomy and performance deficiency schema as a mediator in the relationship between stress-related growth and psychological distress emphasizes the role of cognitive processes in shaping emotional post-trauma responses. This nuanced understanding can inform therapeutic interventions aimed at addressing and restructuring these schemas to promote psychological well-being. From a clinical perspective, our findings have direct implications for trauma-focused interventions. Identifying the mediating role of the autonomy and performance deficiency schema suggests that therapeutic approaches targeting this schema may be particularly effective in reducing psychological distress in road traffic accident participants.

It is important to acknowledge the limitations of our study, including potential sample biases and reliance on self-report measures. Future research may benefit from longitudinal designs to explore the temporal dynamics of posttraumatic growth and cognitive schemas. Additionally, investigating the effectiveness of interventions targeting specific schemas could yield valuable insights.

Furthermore, we acknowledge the limitation of not explicitly addressing the age of the respondents in our analyses. While our research primarily focuses on the interplay between posttraumatic growth, maladaptive cognitive schemas, and psychological distress, we understand that age can be a significant and diverse variable that may influence these relationships. Age-related variations in cognitive and emotional processes are well-established in developmental psychology, and it is possible that such variations could impact the findings of our study. However, given the scope of our research and the specific aims we set out to address, we made the decision to exclude age as a primary variable in our analyses. This choice was based on our commitment to maintaining a clear and concise focus on our research questions.

The implications of this combined understanding are substantial for clinical practice and interventions targeting individuals with psychotraumatic disorders. Mental health professionals can utilize insights from both the COR theory and cognitive-behavioral strategies that challenge and modify maladaptive schemas. By addressing the cognitive schemas’ influence on well-being and the conservation of resources, clinicians can design more comprehensive treatment approaches that alleviate psychological distress, facilitate posttraumatic growth, and ultimately improve the mental health and overall quality of life for those affected by these disorders. Future research should continue to explore these associations, providing valuable insights into the development of effective interventions that can enhance the well-being of individuals recovering from psychotraumatic experiences.

## Figures and Tables

**Figure 1 healthcare-11-02959-f001:**
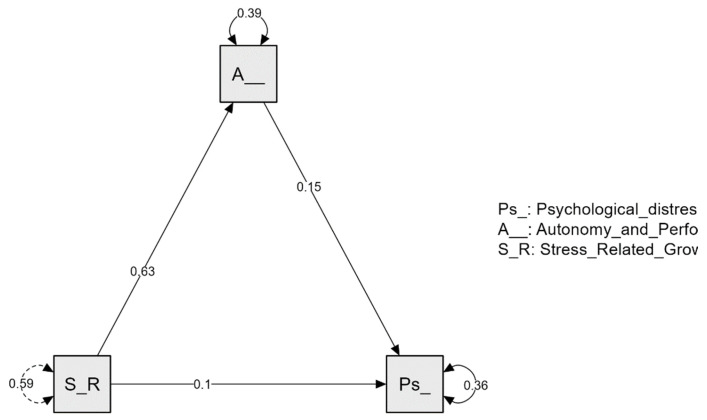
Path plot.

**Table 1 healthcare-11-02959-t001:** Descriptive statistics.

				95% Confidence Interval Mean		
	Mean	Std. Error of Mean		Upper	Lower	Std. Deviation
Stress-Related Growth	4.281	0.033	0.770	4.347	4.215	0.770
Separation and Rejection Schema	4.147	0.037	0.844	4.219	4.075	0.844
Maladaptive Limits Schema	4.422	0.033	0.768	4.488	4.357	0.768
Excessive Other Directedness Schema	4.269	0.037	0.849	4.341	4.197	0.849
Hypervigilance and Inhibition Schema	4.435	0.032	0.744	4.498	4.371	0.744
Autonomy and Performance Deficiency Schema	4.369	0.034	0.791	4.436	4.301	0.791
Psychological Distress	3.719	0.027	0.627	3.773	3.666	0.627

**Table 2 healthcare-11-02959-t002:** Correlation matrix between the research variables.

Variable	1	2	3	4	5	6	7
1. Stress-Related Growth	-						
2. Separation and Rejection Schema	0.802 **	-					
3. Maladaptive Limits Schema	0.690 **	0.688 **	-				
4. Excessive Other Directedness Schema	0.749 **	0.781 **	0.764 **	-			
5. Hypervigilance and Inhibition Schema	0.679 **	0.690 **	0.832 **	0.793 **	-		
6. Autonomy and Performance Deficiency Schema	0.612 **	0.668 **	0.752 **	0.713 **	0.827 **	-	
7. Psychological distress	0.239 **	0.236 **	0.256 **	0.239 **	0.260 **	0.265 **	-

Note: ** *p* < 0.01.

**Table 3 healthcare-11-02959-t003:** Direct effects of stress-related growth on psychological distress.

Direct Effects								
							95% Confidence Interval	
			Estimate	Std. Error	z-Value	*p*	Lower	Upper
Stress-Related Growth	→	Psychological Distress	0.101	0.091	1.109	0.26	−0.078	0.281

Note: Robust standard errors, robust confidence intervals, ML estimator.

**Table 4 healthcare-11-02959-t004:** Indirect effects.

									95% Confidence Interval	
					Estimate	Std. Error	z-Value	*p*	Lower	Upper
Stress-Related Growth	→	Separation and rejection schema	→	Psychological distress	0.024	0.084	0.286	0.77	−0.140	0.188
Stress-Related Growth	→	Maladaptive limits schema	→	Psychological distress	0.058	0.066	0.874	0.38	−0.072	0.187
Stress-Related Growth	→	Excessive other directedness schema	→	Psychological distress	−0.007	0.078	−0.090	0.92	−0.160	0.146
Stress-Related Growth	→	Hypervigilance and inhibition schema	→	Psychological distress	0.033	0.080	0.408	0.68	−0.124	0.189
Stress-Related Growth	→	Autonomy and performance deficiency schema	→	Psychological distress	0.102	0.053	1.929	0.05	−0.002	0.206

Note: Robust standard errors, robust confidence intervals, ML estimator.

## Data Availability

Data will be made available on request by the first author and the corresponding author.

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
