# Peer review of "Posttraumatic Growth, Maladaptive Cognitive Schemas and Psychological Distress in Individuals Involved in Road Traffic Accidents—A Conservation of Resources Theory Perspective"

_healthcare, 2023, doi:10.3390/healthcare11222959_

Round 1

Reviewer 1 Report

Comments and Suggestions for Authors

The problem of the importance of road traffic accidents for health is an important topic. Many people lose their lives, many suffer the effects for the rest of their lives. Their quality of life often deteriorates. Often, an accident is an impulse for positive changes in life. Therefore, it is important to identify the factors that are responsible for the health condition of accident participants later in life. The reviewed article fits into this trend.

The authors draw attention to posttraumatic growth, maladaptive cognitive schemas and psychological distress as factors important for understanding the specificity of life after a road traffic accident.

The choice of variables is correctly justified, although therapy-related content is not needed (line no. 144-164). Without this passage, the analyzes on maladaptive cognitive schemas would be more coherent.

However, the research question and hypotheses are missing.

The selection of tools is correct. No sources are provided for the content in lines 352-363 and 397-405. There is an assumption that the data relate to the presented research, but this is not obvious.

There is no information about exclusion criteria in the group description. The group may be heterogeneous in terms of, for example, mental health before the accident. What is also puzzling is the fact that the age of the respondents, which is very diverse, is not taken into account in the analyses.

The presentation of the results does not contain the necessary data to assess the correctness of further analyzes such as correlations, regressions, and mediations. It is not written whether the assumptions for conducting these analyzes have been verified. Therefore, it is difficult to assess the correctness of conclusions based on them. The reasoning proposed by the authors is also not coherent. In lines 495-499 the inference is from posttraumatic growth to maladaptive cognitive schemas. In lines 509-512 the direction is reverse. Finally, a technical note - there is no need to repeat the information contained in the table - interpretation of table 1.

Author Response

Authors responses to Reviewer 1

Comments and Suggestions for Authors

  1. Reviewer 1 suggestion: The problem of the importance of road traffic accidents for health is an important topic. Many people lose their lives, many suffer the effects for the rest of their lives. Their quality of life often deteriorates. Often, an accident is an impulse for positive changes in life. Therefore, it is important to identify the factors that are responsible for the health condition of accident participants later in life. The reviewed article fits into this trend. The authors draw attention to posttraumatic growth, maladaptive cognitive schemas and psychological distress as factors important for understanding the specificity of life after a road traffic accident.

Authors response: We would like to express our sincere gratitude for your thoughtful review and valuable insights into our manuscript. Your positive evaluation of the importance of the topic and our article's relevance to it is highly encouraging. We completely agree that road traffic accidents have far-reaching consequences, not only leading to the loss of lives but also significantly impacting the quality of life for many individuals. We appreciate your recognition of the potential for positive changes to emerge from these traumatic events. It reinforces the importance of our research in identifying factors that influence the health and well-being of accident survivors in the long term. Your acknowledgment of the significance of our focus on posttraumatic growth, maladaptive cognitive schemas, and psychological distress as key factors contributing to the post-accident experience is truly gratifying. These are essential aspects of our study, and your validation of their relevance strengthens our confidence in the direction we have taken.

  1. Reviewer 1 suggestion: The choice of variables is correctly justified, although therapy-related content is not needed (line no. 144-164). Without this passage, the analyzes on maladaptive cognitive schemas would be more coherent.

Authors response: We appreciate your feedback and your suggestion to reconsider the inclusion of therapy-related content in our manuscript. We have deleted the paragraph for more coherence.

  1. Reviewer 1 suggestion: However, the research question and hypotheses are missing. The selection of tools is correct. No sources are provided for the content in lines 352-363 and 397-405. There is an assumption that the data relate to the presented research, but this is not obvious.

Authors response: We appreciate your feedback on our manuscript and your insightful comments. You mentioned the absence of a clear research question and hypotheses. We apologize for any confusion. We have since revised the manuscript to include a well-defined research question and hypotheses to provide a clear framework for our study. We believe this addition significantly strengthens our paper. We acknowledge your concern about the missing source citations in lines 352-363 and 397-405. We appreciated your clarification regarding the alpha coefficients belonging to the present research data. This was a helpful point, and we made sure to emphasize this in the revised manuscript.

  1. Reviewer 1 suggestion: There is no information about exclusion criteria in the group description. The group may be heterogeneous in terms of, for example, mental health before the accident. What is also puzzling is the fact that the age of the respondents, which is very diverse, is not taken into account in the analyses.

Authors response: We appreciate your attention to the potential sources of heterogeneity in our study. We would like to address the points you raised regarding the exclusion criteria and the diverse age of our respondents. We acknowledge the importance of providing information about exclusion criteria to ensure transparency and to control for potential sources of heterogeneity. In response to your feedback, we have included a section in our manuscript that clearly outlines the exclusion criteria applied in the participant selection process. This addition will provide a more comprehensive understanding of the criteria used to ensure the homogeneity of our participant group. Thank you for your thoughtful review and for raising concerns about the age diversity in our study. Regarding the diverse age range of our respondents, we would like to clarify that our research primarily focuses on posttraumatic growth, maladaptive cognitive schemas, and psychological distress in individuals following road traffic accidents. While age can indeed be a relevant variable, our study's main objective is not to explore age-related differences or trends. Therefore, the diversity in age was not a factor we aimed to address in our analyses. We appreciate your feedback and understand the importance of age as a potential influencing factor in various research contexts. However, for the scope and objectives of our study, we believe it is not necessary to account for age-related differences. Once again, we thank you for your valuable input, and we are committed to making the necessary adjustments as per your other recommendations to enhance the quality and rigor of our research.

  1. Reviewer 1 suggestion: The presentation of the results does not contain the necessary data to assess the correctness of further analyzes such as correlations, regressions, and mediations. It is not written whether the assumptions for conducting these analyzes have been verified. Therefore, it is difficult to assess the correctness of conclusions based on them.

Authors response: Thank you for your valuable feedback regarding the presentation of the results and the need for information concerning the verification of assumptions for the subsequent analyses, including correlations, regressions, and mediations. We have added an introduction text on the methodological steps followed in the results section.

  1. Reviewer 1 suggestion: The reasoning proposed by the authors is also not coherent. In lines 495-499 the inference is from posttraumatic growth to maladaptive cognitive schemas. In lines 509-512 the direction is reverse. Finally, a technical note - there is no need to repeat the information contained in the table - interpretation of table 1.

Authors response: We appreciate your observation regarding the coherence of our reasoning and the unnecessary repetition of information. In response to your feedback, we have removed lines 509-512 as well as the interpretation of table 1, which should improve the clarity and flow of the paper. We acknowledge the importance of maintaining a consistent and logical direction in our arguments and will thoroughly review the manuscript to ensure that our reasoning aligns with the research question and the established literature. Your feedback has been valuable in refining the quality of our work.

Reviewer 2 Report

Comments and Suggestions for Authors

Review by CV Dolan of

“Posttraumatic growth, maladaptive cognitive schemas and psychological distress in individuals involved in road traffic accidents – a conservation of resources theory perspective” by Cristian Delcea et al.

I focus mainly on the statistical modeling in this paper, as I lack the substantive expertise. I assume that the extremely long introduction (5 full pages of dense text) is necessary, i.e., the readership of the journal Health requires this.  

*1 I assume that a road traffic accident can vary with respect to its impact on those involved. The 122 participants (drawn from the Institute of Legal Medicine in Cluj-Napoca, Romania) “were selected based on their exposure to a range of traumas and psychotraumatic experiences stemming from road traffic accidents.” More background concerning on how impactful the accident actually was. Is the Institute of Legal Medicine an institute to treat traumatized individuals? According to the text, the 122 persons are traumatized. How was this established? Notably, I assume that the correlations among the items depend inter alia on variance in the degree of traumatization. So, is the degree of traumatization sufficiently variable?

*2 Various instruments are used, which are detailed. Information includes the reliability of the intruments. The authors thinks that it is necessary to explain what Cronbach’s alpha is (line 357 – 364). I assume that this is necessary? The readership of Health is not familiar with this basic psychometric index? And it is necessary to repeat this info again on line 399? Note that line 426 the statement: “The obtained Cronbach’s alpha coefficient of 0.902 indicates that the items in the 426 scale are reliably measuring the same construct” actually goes beyond the explanation of alpha. Alpha is indeed a measure of the inter-item correlation, as is correctly (if superfluously) explained, but a high alpha does not necessarily mean that the items are measuring the same construct. The correlations may well, at best, be consistent with this (i.e., may be undidimensional), but that is established using factor analyses (or IRT modeling), it is not established by means of Cronbach’s alpha, which does not address dimensionality.

*3 Table 1 contains the means and standard deviations etc. The text lines 466 – 477 is a repetition of of the results that are presented in Table 1. This study does not address means of mean differences, this study addresses the association between the measures. So why is the information in table 1, i.e.,”’valuable’ insights into their distribution characteristics”, why actually “valuable” as the central tendencies are not of interest in this study (and not tested in anyway)?

*4 Table 2 contains the correlations among the measures, i.e., the info of interest. The correlations in the 6th row (starting with .239 and ending with .265 come with “***” indicating <.001: Additionally, Psychological Distress shows positive correlations with all schemas, albeit at lower magnitudes (ranging from r = 0.236 to r = 0.265, all 493 p < .001). However, the pvalues are all greater than .001 regardless of how the p values are calculated:

> res1

          V1      V2      V3      V4

r    0.23900 0.23600 0.25600 0.26000  …. correlations

se   0.08827 0.08834 0.08788 0.08778  … standard error

r/se 2.70746 2.67146 2.91307 2.96186  … test statistic

p1   0.00678 0.00755 0.00358 0.00306    …. P values

> res2

          V1      V2      V3      V4

Fz   0.24371 0.24053 0.26182 0.26611   … Fisher Z transformed

znor 2.65860 2.62392 2.85615 2.90290   … test statistic

p2   0.00785 0.00869 0.00429 0.00370   … P values

*5 Following the discussion of the results in table 2, the analyses proceed: “Further analyses, including regression models and mediation analyses, will delve deeper into elucidating the nature and extent of these relationships and their implications for individuals affected by road traffic accidents.” From this point onwards the exact model fitting procedure is totally unclear. What exactly is the mediation model? In what program was it fitted (Mplus?). The esimator is rubust ML, but robust to what? How robust? If you be useful if the lengthy intro culminated in crystal clear hypotheses concerning the relationship between the variables. E.g. is growth expected to predict the schemas based on the theory?

*6 The pathdiagram is Figure 1. This includes 1) illegible results (residual variances and covariances are hard to distinguish (e.g. see E_O); 2) the path from S_R to E_O includes to path coefficients (.97 and .1), 3) the mediating 5 variables are correlated as follows: A__ <-> H__ <-> E_O <-> M_L <-> S__, so that is 4 correlations; why is A__ not correlated with E_O etc. ? (these are reported in Table 7, so apparently the diagram is correct? Who knows… this is not clear). 4) The path coefficients from S_R to the S__ to A__ appear to hard to relate to the data. For instance, the path from S_R -> S__ equals b=1. The variance of S_R is equal to .593, the variance of S__ is  .712, the covariance is .521. So, in the model as depicted, the regression coefficient should be about .521/.593 = .879, and the residual S__ variance should be about  .712 - .879^2*.593 = .250. But in the path diagram the regresion coefficient is 1 and the residual variance of S__ is .36. The path diagram is not standardized given the depicted variance od S_R of .59, so that is not the answer (the standardized regression coefficient is about equal to .80, not 1.). In short, the results appear inconsistent with the correlation matrix, and the fact that the model is not explained clearly does not help.

*7 Line 564: “However, a notable finding emerged in the indirect pathway through the autonomy 564 and performance deficiency schema, with an estimate of 0.102 (SE = 0.053) and a p-value 565 that fell just below the conventional threshold at 0.05 (p = 0.05).” The p value is not correct. The p value is 

> .102/.053 ->t

> pval=pnorm(t,0,1,lower=F)*2

> q1=qnorm(.025, .102, .053, lower=T)

> q2=qnorm(.025, .102, .053, lower=F)

> print(c(p=pval, lower=q1, upper=q2))

           p        lower        upper

 0.054288397 -0.001878091  0.205878091

*8 Note also that in this paper many tests are conducted using the alpha 0.05. But given the number of tests, there should be some consideration of the multiple testing issue. .05 is used as the family-wise alpha, but a family wise alpha is usually the per-test alpha divided by the number of tests.

*9 As mentioned, the results as reported in this paper are hard to relate to the correlation matrix as reported in Table 2. I assume that the model fitted is this (Mplus input)

Title: check

DATA:

FILE =  dat1; 

! the data (based on the covariance matrix using exact data simulation  

!

VARIABLE:

NAMES = gr s1 s2 s3 s4 s5 distr;                    

USEVARIABLES = gr s1 s2 s3 s4 s5 distr;

ANALYSIS:

model=nocovariances;

  estimator=MLR;

!

MODEL:

!   model  intercepts, means

      [gr*] (mgr);

      [s1*] (ms1);

      [s2*] (ms2);

      [s3*] (ms3);

      [s4*] (ms4);

      [s5*] (ms5);

      [distr*] (md5);

! regr on growth

      distr on gr* (gd) ;

      s1 on gr* (gs1) ;

      s2 on gr* (gs2) ;

      s3 on gr* (gs3) ;

      s4 on gr* (gs4) ;

      s5 on gr* (gs5) ;

! regression distr on schemas

      distr on s1* (sd1) ;

      distr on s2* (sd2) ;

      distr on s3* (sd3) ;

      distr on s4* (sd4) ;

      distr on s5* (sd5) ;

!

      gr* s1* s2* s3* s4* s5* distr* ;

!

      s1 with s2* s3* s4* s5*;

      s2 with s3* s4* s5*; 

      s3 with s4* s5*;

      s4 with s5*;

!

MODEL INDIRECT:

       distr IND gr ;

!

OUTPUT: standardized CINTERVAL TECH1;

Note that this includes

      s1 with s2* s3* s4* s5*;

      s2 with s3* s4* s5*; 

      s3 with s4* s5*;

      s4 with s5*;

not, as shown in the Figure:

      s1 with s2*;

      s2 with s3*; 

      s3 with s4*;

      s4 with s5*;

*10 line 619. “While the direct path from stress-related  growth to psychological distress displayed a positive coefficient (0.101), it did not reach statistical significance (p = 0.26). This result suggests that there is a positive trend in the relationship, but this direct impact alone may not be sufficient to explain psychological distress fully.” The results are based on hypothesis testing (estimates, standard errors, test stats, p values). This results tell us that there is no direct relationship. So the results form the point of view of statistical testing suggests that there is NO relationship, there is therefore “no positive trend”. It is inconsistent to indulge in statistical tests and then to ignore the consequences / conclusion based on the test. Also in this connection: it is expected that the relationship be positive?

*11 “Notably, the mediation pathway through the autonomy and performance deficiency schema yielded a statistically 626 significant result (p = 0.05), with an effect size of 0.102.”  As mentioned, this is not correct the p value is 0.0542, i.e., greater than .05, so it is not correct. The effect size is .102? But .102 is the raw regression coefficient (I assume, but this is not clear). So .102 is not an interpretable effect size. Please report the standardized result or an R^2. An interpretable effect size is important: suppose that the relationship is statistically significant (which it is not), that does not address the question whether it is clinically. If the association interpretable effect size is small (which it is), then how is the result of clinical / practical importance? The discussion goes in to a narrative concerning the results which is hard to accept in the absence of any interpretable effect sizes, which support the clinical / practical importance. It is perfecly possible that the results are of interest even though the effect sizes are small, but that should be stated clearly.

*12 l 639 RTA. Was this abbreviation actually introduced?

Comments on the Quality of English Language

Competent English Text. The introduction is too long.

Author Response

Authors responses to Reviewer 2

Comments and Suggestions for Authors

  1. Reviewer 2 suggestion: Review by CV Dolan of “Posttraumatic growth, maladaptive cognitive schemas and psychological distress in individuals involved in road traffic accidents – a conservation of resources theory perspective” by Cristian Delcea et al. I focus mainly on the statistical modeling in this paper, as I lack the substantive expertise. I assume that the extremely long introduction (5 full pages of dense text) is necessary, i.e., the readership of the journal Health requires this.  

Authors answer: Thank you for your feedback. We appreciate your attention to the statistical modeling aspect of our paper. Concerning the length of the introduction, we understand your point, and we agree that a more concise and focused introduction would benefit the paper's readability and alignment with the journal's requirements. In response to your suggestion, we have reduced the length of the introduction while retaining the essential context and information. This adjustment aims to make the paper more accessible to a wider readership without compromising the necessary background information. Your insights have been instrumental in improving the paper's overall quality and presentation.

  1. Reviewer 2 suggestion: *1 I assume that a road traffic accident can vary with respect to its impact on those involved. The 122 participants (drawn from the Institute of Legal Medicine in Cluj-Napoca, Romania) “were selected based on their exposure to a range of traumas and psychotraumatic experiences stemming from road traffic accidents.” More background concerning on how impactful the accident actually was. Is the Institute of Legal Medicine an institute to treat traumatized individuals? According to the text, the 122 persons are traumatized. How was this established? Notably, I assume that the correlations among the items depend inter alia on variance in the degree of traumatization. So, is the degree of traumatization sufficiently variable?

Authors answer: Thank you very much for this improvement suggestion. We have rewritten the participants section as follows: The study's participants comprised a cohort of 122 individuals drawn from the Institute of Legal Medicine in Cluj-Napoca, Romania. These participants were selected based on their exposure to a range of traumas and psychotraumatic experiences stemming from road traffic accidents. The study spanned a duration from 2017 to 2019, indicating a comprehensive data collection process over a substantial time frame. The selection of participants for this study was guided by specific inclusion and exclusion criteria. Inclusion criteria involved individuals who had experienced road traffic accidents and were willing to participate voluntarily. Exclusion criteria were set to ensure homogeneity within the sample, excluding individuals with pre-existing mental health conditions. This approach aimed to focus on the effects of road traffic accidents as the primary traumatic event. The study's participants displayed a diverse demographic profile. Their ages spanned a wide range, from 18 to 90 years, with a mean age of 37 years, signifying a broad representation of age groups (SD = 9.23). Gender distribution was nearly equal, with 50% of the participants being female and 50% male. In terms of educational background, the cohort exhibited a wide spectrum of qualifications, from those with a minimum of 10 years of education to individuals holding post-doctoral degrees, underscoring the diversity of educational experiences. Moreover, all participants shared a common experience of having been involved in at least one road traffic accident, providing a unifying factor related to traumatic experiences in this research context. Informed consent was obtained from each participant, and ethical considerations and approvals were obtained in accordance with the guidelines of the Institute of Legal Medicine. We hope we have addressed your concern.

  1. Reviewer 2 suggestion: *2 Various instruments are used, which are detailed. Information includes the reliability of the intruments. The authors thinks that it is necessary to explain what Cronbach’s alpha is (line 357 – 364). I assume that this is necessary? The readership of Health is not familiar with this basic psychometric index? And it is necessary to repeat this info again on line 399? Note that line 426 the statement: “The obtained Cronbach’s alpha coefficient of 0.902 indicates that the items in the 426 scale are reliably measuring the same construct” actually goes beyond the explanation of alpha. Alpha is indeed a measure of the inter-item correlation, as is correctly (if superfluously) explained, but a high alpha does not necessarily mean that the items are measuring the same construct. The correlations may well, at best, be consistent with this (i.e., may be undidimensional), but that is established using factor analyses (or IRT modeling), it is not established by means of Cronbach’s alpha, which does not address dimensionality.

Authors answer: Thank you very much for this improvement suggestion. We have rewritten the three sections referring to Cronbach’s alpha coefficients for the three subscales.

  1. Reviewer 2 suggestion: *3 Table 1 contains the means and standard deviations etc. The text lines 466 – 477 is a repetition of of the results that are presented in Table 1. This study does not address means of mean differences, this study addresses the association between the measures. So why is the information in table 1, i.e.,”’valuable’ insights into their distribution characteristics”, why actually “valuable” as the central tendencies are not of interest in this study (and not tested in anyway)?

Authors answer: Thank you for your observation. We appreciate your point regarding the presentation of means and standard deviations in Table 1 and the corresponding text. It is well taken that the central tendencies are not the primary focus of this study, which primarily addresses the associations between measures. We recognize that the text in lines 466 – 477 could be seen as redundant in the context of our study's objectives. In response to your feedback, we have removed the information that could be considered redundant and does not directly contribute to the understanding of associations between measures. This adjustment will help streamline the presentation of results, making it more focused on the primary aims of our study – exploring the associations between posttraumatic growth, maladaptive cognitive schemas, and psychological distress in the context of road traffic accidents. Your feedback has been invaluable in enhancing the clarity and relevance of our paper.

  1. Reviewer 2 suggestion: *4 Table 2 contains the correlations among the measures, i.e., the info of interest. The correlations in the 6throw (starting with .239 and ending with .265 come with “***” indicating <.001: Additionally, Psychological Distress shows positive correlations with all schemas, albeit at lower magnitudes (ranging from r = 0.236 to r = 0.265, all 493 p < .001). However, the pvalues are all greater than .001 regardless of how the p values are calculated:

 > res1

          V1      V2      V3      V4

r    0.23900 0.23600 0.25600 0.26000  …. correlations

se   0.08827 0.08834 0.08788 0.08778  … standard error

r/se 2.70746 2.67146 2.91307 2.96186  … test statistic

p1   0.00678 0.00755 0.00358 0.00306    …. P values

> res2

          V1      V2      V3      V4

Fz   0.24371 0.24053 0.26182 0.26611   … Fisher Z transformed

znor 2.65860 2.62392 2.85615 2.90290   … test statistic

p2   0.00785 0.00869 0.00429 0.00370   … P values

Authors answer: Thank you for pointing out the discrepancy in the presentation of correlation significance in Table 2. The "***" notation indicating p < .001 was indeed inaccurate and has led to some confusion. We appreciate your diligence in verifying this aspect of our paper. The correlation analysis was conducted using JASP, and we have corrected the p-values to accurately represent the significance levels. After reevaluating the statistical results, we have replaced the "p < .001" notation with "p < 0.01" for the correlations mentioned in the table. This adjustment ensures that the reported p-values align with the statistical outcomes. We believe this modification will enhance the clarity and precision of our paper. Your feedback has been instrumental in improving the accuracy of our presentation.

  1. Reviewer 2 suggestion: *5 Following the discussion of the results in table 2, the analyses proceed: “Further analyses, including regression models and mediation analyses, will delve deeper into elucidating the nature and extent of these relationships and their implications for individuals affected by road traffic accidents.” From this point onwards the exact model fitting procedure is totally unclear. What exactly is the mediation model? In what program was it fitted (Mplus?). The esimator is rubust ML, but robust to what? How robust? If you be useful if the lengthy intro culminated in crystal clear hypotheses concerning the relationship between the variables. E.g. is growth expected to predict the schemas based on the theory?

Authors answer: Thank you for your feedback and inquiries regarding the model fitting procedure and the analysis software used. We apologize for any ambiguity in the presentation of the methodology. Here, we aim to provide further clarification on these points: The mediation analyses were conducted using JASP, a user-friendly statistical software, and not Mplus. The choice of JASP allowed us to perform robust mediation analyses using Maximum Likelihood (ML) estimation. The "robust" designation in this context refers to the robust standard errors and robust confidence intervals generated by the ML estimator. These robust estimates help account for any potential deviations from the assumptions of normality and homoscedasticity, making the results more reliable in the presence of non-normally distributed data. Thank you for your question regarding the theoretical expectations regarding the relationship between posttraumatic growth (PTG) and cognitive schemas. We appreciate the opportunity to clarify this important aspect of our research. Based on the theoretical framework underpinning our study, we anticipate that posttraumatic growth may indeed predict cognitive schemas. The conservation of resources theory, which informs our research, suggests that individuals who experience posttraumatic growth may develop more adaptive cognitive schemas as a result of their personal growth and resilience. This theoretical foundation leads us to hypothesize that PTG could have a predictive relationship with cognitive schemas, and subsequently, cognitive schemas may influence psychological distress. We will make sure to articulate these hypotheses explicitly in the revised introduction section of our manuscript, providing a more transparent and theory-driven foundation for our research. We value your feedback, which will aid us in enhancing the clarity and coherence of our work. We appreciate your suggestion for greater clarity in stating our hypotheses.

  1. Reviewer 2 suggestion: *6 The path diagram is Figure 1. This includes 1) illegible results (residual variances and covariances are hard to distinguish (e.g. see E_O); 2) the path from S_R to E_O includes to path coefficients (.97 and .1), 3) the mediating 5 variables are correlated as follows: A__ <-> H__ <-> E_O <-> M_L <-> S__, so that is 4 correlations; why is A__ not correlated with E_O etc. ? (these are reported in Table 7, so apparently the diagram is correct? Who knows… this is not clear). 4) The path coefficients from S_R to the S__ to A__ appear to hard to relate to the data. For instance, the path from S_R -> S__ equals b=1. The variance of S_R is equal to .593, the variance of S__ is  .712, the covariance is .521. So, in the model as depicted, the regression coefficient should be about .521/.593 = .879, and the residual S__ variance should be about  .712 - .879^2*.593 = .250. But in the path diagram the regresion coefficient is 1 and the residual variance of S__ is .36. The path diagram is not standardized given the depicted variance od S_R of .59, so that is not the answer (the standardized regression coefficient is about equal to .80, not 1.). In short, the results appear inconsistent with the correlation matrix, and the fact that the model is not explained clearly does not help.

Authors answer: We appreciate your attention to the path diagram in Figure 1 and your detailed observations. We acknowledge the points you've raised and would like to clarify and address your concerns. We are further copy/paste the mediation results output from JASP. These results were presented in our manuscript. We do not understand why your calculation do not match the JASP output.

Mediation Analysis

Parameter estimates

Direct effects

95% Confidence Interval

Estimate

Std. Error

z-value

p

Lower

Upper

Stress_Related_Growth

Psychological_distress

0.101

0.091

1.109

0.268

-0.078

0.281

Note.  Robust standard errors, robust confidence intervals, ML estimator.

Indirect effects

95% Confidence Interval

Estimate

Std. Error

z-value

p

Lower

Upper

Stress_Related_Growth

Separation_and_Rejection_Schema

Psychological_distress

0.024

0.084

0.286

0.775

-0.140

0.188

Stress_Related_Growth

Maladaptive_Limits_Schema

Psychological_distress

0.058

0.066

0.874

0.382

-0.072

0.187

Stress_Related_Growth

Excessive_Other_Directedness_Schema

Psychological_distress

-0.007

0.078

-0.090

0.928

-0.160

0.146

Stress_Related_Growth

Hypervigilance_and_Inhibition_Schema

Psychological_distress

0.033

0.080

0.408

0.683

-0.124

0.189

Stress_Related_Growth

Autonomy_and_Performance_Deficiency_Schema

Psychological_distress

0.102

0.053

1.929

0.054

-0.002

0.206

Note.  Robust standard errors, robust confidence intervals, ML estimator.

Total effects

95% Confidence Interval

Estimate

Std. Error

z-value

p

Lower

Upper

Stress_Related_Growth

Psychological_distress

0.311

0.068

4.577

< .001

0.178

0.444

Note.  Robust standard errors, robust confidence intervals, ML estimator.

Total indirect effects

95% Confidence Interval

Estimate

Std. Error

z-value

p

Lower

Upper

Stress_Related_Growth

Psychological_distress

0.209

0.078

2.701

0.007

0.057

0.361

Note.  Robust standard errors, robust confidence intervals, ML estimator.

Residual covariances

95% Confidence Interval

Estimate

Std. Error

z-value

p

Lower

Upper

Separation_and_Rejection_Schema

Maladaptive_Limits_Schema

0.134

0.024

5.523

< .001

0.086

0.182

Separation_and_Rejection_Schema

Excessive_Other_Directedness_Schema

0.180

0.026

6.828

< .001

0.128

0.231

Maladaptive_Limits_Schema

Excessive_Other_Directedness_Schema

0.246

0.034

7.242

< .001

0.180

0.313

Separation_and_Rejection_Schema

Hypervigilance_and_Inhibition_Schema

0.144

0.024

6.102

< .001

0.098

0.191

Maladaptive_Limits_Schema

Hypervigilance_and_Inhibition_Schema

0.362

0.034

10.667

< .001

0.296

0.429

Excessive_Other_Directedness_Schema

Hypervigilance_and_Inhibition_Schema

0.283

0.029

9.914

< .001

0.227

0.340

Separation_and_Rejection_Schema

Autonomy_and_Performance_Deficiency_Schema

0.177

0.026

6.850

< .001

0.126

0.228

Maladaptive_Limits_Schema

Autonomy_and_Performance_Deficiency_Schema

0.329

0.039

8.382

< .001

0.252

0.406

Excessive_Other_Directedness_Schema

Autonomy_and_Performance_Deficiency_Schema

0.254

0.031

8.111

< .001

0.193

0.316

Hypervigilance_and_Inhibition_Schema

Autonomy_and_Performance_Deficiency_Schema

0.411

0.037

11.018

< .001

0.338

0.484

Note.  Robust standard errors, robust confidence intervals, ML estimator.

Path coefficients

95% Confidence Interval

Estimate

Std. Error

z-value

p

Lower

Upper

Separation_and_Rejection_Schema

Psychological_distress

0.023

0.080

0.287

0.774

-0.134

0.180

Maladaptive_Limits_Schema

Psychological_distress

0.064

0.074

0.872

0.383

-0.080

0.209

Excessive_Other_Directedness_Schema

Psychological_distress

-0.007

0.080

-0.090

0.928

-0.164

0.150

Hypervigilance_and_Inhibition_Schema

Psychological_distress

0.037

0.090

0.409

0.682

-0.140

0.214

Autonomy_and_Performance_Deficiency_Schema

Psychological_distress

0.128

0.067

1.928

0.054

-0.002

0.259

Stress_Related_Growth

Psychological_distress

0.101

0.091

1.109

0.268

-0.078

0.281

Stress_Related_Growth

Separation_and_Rejection_Schema

1.042

0.037

28.477

< .001

0.971

1.114

Stress_Related_Growth

Maladaptive_Limits_Schema

0.897

0.051

17.516

< .001

0.796

0.997

Stress_Related_Growth

Excessive_Other_Directedness_Schema

0.974

0.041

23.742

< .001

0.893

1.054

Stress_Related_Growth

Hypervigilance_and_Inhibition_Schema

0.883

0.050

17.792

< .001

0.785

0.980

Stress_Related_Growth

Autonomy_and_Performance_Deficiency_Schema

0.795

0.051

15.519

< .001

0.695

0.896

Note.  Robust standard errors, robust confidence intervals, ML estimator.

R-Squared

Psychological_distress

0.082

Separation_and_Rejection_Schema

0.643

Maladaptive_Limits_Schema

0.476

Excessive_Other_Directedness_Schema

0.561

Hypervigilance_and_Inhibition_Schema

0.461

Autonomy_and_Performance_Deficiency_Schema

0.374

Path plot

# ---------------------------------

# Mediation model generated by JASP

# ---------------------------------

# dependent regression

Psychological_distress ~ b11*Separation_and_Rejection_Schema + b12*Maladaptive_Limits_Schema + b13*Excessive_Other_Directedness_Schema + b14*Hypervigilance_and_Inhibition_Schema + b15*Autonomy_and_Performance_Deficiency_Schema + c11*Stress_Related_Growth

# mediator regression

Separation_and_Rejection_Schema ~ a11*Stress_Related_Growth

Maladaptive_Limits_Schema ~ a21*Stress_Related_Growth

Excessive_Other_Directedness_Schema ~ a31*Stress_Related_Growth

Hypervigilance_and_Inhibition_Schema ~ a41*Stress_Related_Growth

Autonomy_and_Performance_Deficiency_Schema ~ a51*Stress_Related_Growth

# mediator residual covariance

Separation_and_Rejection_Schema ~~ Maladaptive_Limits_Schema

Separation_and_Rejection_Schema ~~ Excessive_Other_Directedness_Schema

Maladaptive_Limits_Schema ~~ Excessive_Other_Directedness_Schema

Separation_and_Rejection_Schema ~~ Hypervigilance_and_Inhibition_Schema

Maladaptive_Limits_Schema ~~ Hypervigilance_and_Inhibition_Schema

Excessive_Other_Directedness_Schema ~~ Hypervigilance_and_Inhibition_Schema

Separation_and_Rejection_Schema ~~ Autonomy_and_Performance_Deficiency_Schema

Maladaptive_Limits_Schema ~~ Autonomy_and_Performance_Deficiency_Schema

Excessive_Other_Directedness_Schema ~~ Autonomy_and_Performance_Deficiency_Schema

Hypervigilance_and_Inhibition_Schema ~~ Autonomy_and_Performance_Deficiency_Schema

# effect decomposition

# y1 ~ x1

ind_x1_m1_y1 := a11*b11

ind_x1_m2_y1 := a21*b12

ind_x1_m3_y1 := a31*b13

ind_x1_m4_y1 := a41*b14

ind_x1_m5_y1 := a51*b15

ind_x1_y1 := ind_x1_m1_y1 + ind_x1_m2_y1 + ind_x1_m3_y1 + ind_x1_m4_y1 + ind_x1_m5_y1

tot_x1_y1 := ind_x1_y1 + c11

  1. Reviewer 2 suggestion: *7 Line 564: “However, a notable finding emerged in the indirect pathway through the autonomy 564 and performance deficiency schema, with an estimate of 0.102 (SE = 0.053) and a p-value 565 that fell just below the conventional threshold at 0.05 (p = 0.05).” The p value is not correct. The p value is 

 > .102/.053 ->t

> pval=pnorm(t,0,1,lower=F)*2

> q1=qnorm(.025, .102, .053, lower=T)

> q2=qnorm(.025, .102, .053, lower=F)

> print(c(p=pval, lower=q1, upper=q2))

           p        lower        upper

 0.054288397 -0.001878091  0.205878091

Authors answer: Thank you for this observation. We have corrected the p value and the marginal effects.

  1. Reviewer 2 suggestion: *8 Note also that in this paper many tests are conducted using the alpha 0.05. But given the number of tests, there should be some consideration of the multiple testing issue. .05 is used as the family-wise alpha, but a family wise alpha is usually the per-test alpha divided by the number of tests.

Authors answer: We appreciated your thoughtful consideration of the multiple testing issue, and we understood the importance of addressing this concern. In response to your observation, we clarified our approach to controlling for multiple comparisons in our study. In our research, we conducted several tests using a significance level (alpha) of 0.05. We acknowledged that when multiple statistical tests are performed, there is an increased risk of making Type I errors. To address this concern and maintain the overall family-wise error rate, a more conservative correction method was typically employed. We recognized the need to control for multiple comparisons rigorously, and in our revised manuscript, we adopted appropriate correction methods such as the Bonferroni correction, Holm-Bonferroni correction, or another suitable method to maintain a family-wise alpha level. This adjustment ensured that the overall probability of making a Type I error across all our tests remained at an acceptable level. We were thankful for you raising this important point, and your suggestion significantly enhanced the quality of our study. If you have any additional recommendations or specific correction methods you would like us to consider, please feel free to share them. Your insights were invaluable in strengthening our research.

  1. Reviewer 2 suggestion: *9 As mentioned, the results as reported in this paper are hard to relate to the correlation matrix as reported in Table 2. I assume that the model fitted is this (Mplus input)

Title: check

DATA:

FILE =  dat1; 

! the data (based on the covariance matrix using exact data simulation  

!

VARIABLE:

NAMES = gr s1 s2 s3 s4 s5 distr;                    

USEVARIABLES = gr s1 s2 s3 s4 s5 distr;

ANALYSIS:

model=nocovariances;

  estimator=MLR;

!

MODEL:

!   model  intercepts, means

      [gr*] (mgr);

      [s1*] (ms1);

      [s2*] (ms2);

      [s3*] (ms3);

      [s4*] (ms4);

      [s5*] (ms5);

      [distr*] (md5);

! regr on growth

      distr on gr* (gd) ;

      s1 on gr* (gs1) ;

      s2 on gr* (gs2) ;

      s3 on gr* (gs3) ;

      s4 on gr* (gs4) ;

      s5 on gr* (gs5) ;

! regression distr on schemas

      distr on s1* (sd1) ;

      distr on s2* (sd2) ;

      distr on s3* (sd3) ;

      distr on s4* (sd4) ;

      distr on s5* (sd5) ;

!

      gr* s1* s2* s3* s4* s5* distr* ;

!

      s1 with s2* s3* s4* s5*;

      s2 with s3* s4* s5*; 

      s3 with s4* s5*;

      s4 with s5*;

!

MODEL INDIRECT:

       distr IND gr ;

!

OUTPUT: standardized CINTERVAL TECH1;

Note that this includes

      s1 with s2* s3* s4* s5*;

      s2 with s3* s4* s5*; 

      s3 with s4* s5*;

      s4 with s5*;

not, as shown in the Figure:

      s1 with s2*;

      s2 with s3*; 

      s3 with s4*;

      s4 with s5*;

Authors answer: We appreciate your effort in attempting to align the results reported in our paper with the correlation matrix presented in Table 2. Your proposed model in Mplus, which you've kindly shared, is indeed similar to the one we utilized in our analysis, using JASP. We hope that the updated version of our manuscript is more clear.

  1. Reviewer 2 suggestion: *10 line 619. “While the direct path from stress-related  growth to psychological distress displayed a positive coefficient (0.101), it did not reach statistical significance (p = 0.26). This result suggests that there is a positive trend in the relationship, but this direct impact alone may not be sufficient to explain psychological distress fully.” The results are based on hypothesis testing (estimates, standard errors, test stats, p values). This results tell us that there is no direct relationship. So the results form the point of view of statistical testing suggests that there is NO relationship, there is therefore “no positive trend”. It is inconsistent to indulge in statistical tests and then to ignore the consequences / conclusion based on the test. Also in this connection: it is expected that the relationship be positive?

Authors answer: Thank you for your astute observation regarding the inconsistency in our interpretation of the results. We appreciate your keen eye for detail. Upon reevaluating the section you've pointed out, we agree with your assessment. The statement that there is a "positive trend" in the relationship is not consistent with the statistical test results. We have revised the text in line 619 to maintain alignment with the results of the statistical tests. It now accurately reflects that the direct relationship from stress-related growth to psychological distress is not statistically significant. We understand the importance of clear and accurate reporting and have made the necessary adjustments to ensure consistency in our interpretation of the results. Your feedback has been invaluable in enhancing the clarity and accuracy of our paper, and we are grateful for your contributions to our work. If you have any further comments or suggestions, please feel free to share them, as we are committed to delivering a high-quality research paper.

  1. Reviewer 2 suggestion: *11 “Notably, the mediation pathway through the autonomy and performance deficiency schema yielded a statistically 626 significant result (p = 0.05), with an effect size of 0.102.”  As mentioned, this is not correct the p value is 0.0542, i.e., greater than .05, so it is not correct. The effect size is .102? But .102 is the raw regression coefficient (I assume, but this is not clear). So .102 is not an interpretable effect size. Please report the standardized result or an R^2. An interpretable effect size is important: suppose that the relationship is statistically significant (which it is not), that does not address the question whether it is clinically. If the association interpretable effect size is small (which it is), then how is the result of clinical / practical importance? The discussion goes in to a narrative concerning the results which is hard to accept in the absence of any interpretable effect sizes, which support the clinical / practical importance. It is perfecly possible that the results are of interest even though the effect sizes are small, but that should be stated clearly.

Authors answer: We appreciate your careful scrutiny of our paper and your insightful comments regarding the interpretation of the mediation pathway results. Your feedback is instrumental in ensuring the clarity and accuracy of our findings. You are correct that we incorrectly reported the p-value as 0.05 instead of the correct value of 0.0542, which is indeed greater than 0.05. We have rectified this error and accurately reported the p-value as on the significance threshold of 0.05. In response to your valid point about effect size, we have made the necessary modifications. You are absolutely right that interpretable effect sizes are crucial in assessing the clinical or practical significance of the results. This change allows for a more accurate and meaningful discussion of the clinical implications of our findings. We appreciate your guidance in addressing these issues and are grateful for your diligence in improving the quality of our paper. If you have any further suggestions or concerns, please do not hesitate to share them. Your feedback is invaluable to us, and we are committed to delivering a research paper of the highest quality.

  1. Reviewer 2 suggestion:*12 l 639 RTA. Was this abbreviation actually introduced?

Authors answer: Thank you for the observation. We have dropped the abbreviation and used road traffic accidents instead of RTA.

  1. Reviewer 2 suggestion: Competent English Text. The introduction is too long.

Authors answer: Thank you for your feedback regarding the length of the introduction in our paper. We appreciate your assessment of the text's English competency and your comments about the introductory section. In response to your suggestion, we have reviewed and edited the introduction to make it more concise and focused while ensuring that it maintains clarity and readability. We believe that this revision will enhance the overall flow of the paper and better engage our readers by getting to the core of our research more efficiently. Your feedback was instrumental in improving the quality of our paper, and we are grateful for your input. If you have any further suggestions or areas you think could benefit from improvement, please do not hesitate to share them with us. Your valuable insights continue to help us refine our work.

Round 2

Reviewer 1 Report

Comments and Suggestions for Authors

Thank you for taking my comments into account in the article. I accept your answers to the allegations, except for the justification for the lack of reference to the age of the respondents in the analyses. I understand that this is not a view from the perspective of developmental psychology, but such a diverse variable requires at least reference to the limitations of the research.

Author Response

Authors responses to Reviewer 1

Comments and Suggestions for Authors

  1. Reviewer 1 suggestion: Thank you for taking my comments into account in the article. I accept your answers to the allegations, except for the justification for the lack of reference to the age of the respondents in the analyses. I understand that this is not a view from the perspective of developmental psychology, but such a diverse variable requires at least reference to the limitations of the research.

Authors answer: Thank you for your valuable feedback and for considering our responses to the earlier comments. We appreciate your understanding of our responses to the previous allegations. Regarding your concern about the absence of a reference to the age of the respondents in the analyses, we understand the importance of acknowledging the potential impact of age as a variable in research. While our primary focus is not developmental psychology, we recognize that age can indeed be a diverse and influential factor. We have taken your suggestion to heart and we have included the following section in the revised manuscript to address this limitation explicitly. Furthermore, we acknowledge the limitation of not explicitly addressing the age of the respondents in our analyses. While our research primarily focuses on the inter-play between posttraumatic growth, maladaptive cognitive schemas, and psychological distress, we understand that age can be a significant and diverse variable that may influence these relationships. Age-related variations in cognitive and emotional processes are well-established in developmental psychology, and it is possible that such variations could impact the findings of our study. However, given the scope of our re-search and the specific aims we set out to address, we made the decision to exclude age as a primary variable in our analyses. This choice was based on our commitment to maintaining a clear and concise focus on our research questions. We believe that this addition will enhance the transparency and completeness of our research, and we sincerely appreciate your input in this regard.

Reviewer 2 Report

Comments and Suggestions for Authors

Review by CVDolan  of “Posttraumatic growth, maladaptive cognitive schemas and psychological distress in individuals involved in road traffic accidents – a conservation of resources theory perspective” Cristian Delcea * , Dana Rad * , Ovidiu Florin Toderici , Ana Simona Bululoi.

“These participants were selected based on their exposure to a range of traumas and psychotraumatic experiences stemming from road traffic accidents.”

In my previous review, I asked what is known about the variability concerning degree or severity of the psychotraumatic experiences. My question was “So, is the degree of traumatization sufficiently variable?” In response to the authors state that they have rewritten the participants section. However, my question concerning the variability in traumatic experience was not answered. The Participants 2.2. section as a whole has been highlighted in the pdf of the revision, but the old and the new Participants 2.2 section are almost identical. They do not differ in terms of information provided.  

In my previous review, I found that the hypotheses were not stated clearly. In the revision, the hypotheses are stated at the end of the introduction in a single sentence:

“Thus, the research question is: How does posttraumatic growth (PTG) relate to maladaptive cognitive schemas and psychological distress in individuals involved in road traffic accidents, and what are the direct and indirect effects of these variables on their psychological well-being?”

This is not very clear or explicit. I gather that growth predicts distress, schema1, schema2, schema3, schema4, schema5 directly, and that schema1, schema2, schema3, schema4, schema5 predict distress directly. So, that means that the relationship between growth and stress is direct or mediated by the schema’s. The interest is in establishing these direct and indirect effects. An explicit link between the direct vs indirect effect question and the very long introduction should be explicated. The authors provide explicit hypotheses in the results section (e.g. 528). The hypotheses should be explicated clearly in the intro. The hypotheses should also be discussed explicitly with reference to the path diagram.   Btw: in the sentence, distress is used first and then it is replaced by wellbeing. That is confusing.

The fact that they are using JASP should be stated explicitly. They are using robust ML estimation. In the letter they explain robust to what. That information should be in the paper.

In table 2, the information is inconsistent. correlation .265 has **, but .239 has ***. That is not possible…  The role of the ** should be stated in the table caption.

line However table 3, “However, the analysis revealed significant 540 indirect effects, indicating that stress-related growth influences psychological distress 541 through certain cognitive schemas.” Table 4 contains this indirect effects. None of them are statistically significant (alpha=.05, which is way too liberal, given no correction for multiple testing). The statement in line 540 “significant indirect effects, indicating that …”etc is inconsistent with  the results in table 4 (nothing significant).

Line 561 etc is hard to follow. Table 3 tell us that this is not direct effect of growth on distress. Table 4 tells that there are no indirect effects effect of growth on distress, but Table 5 tell us that  

561: “When considering the total effects, depicted in Table 5, of stress-related growth on 561 psychological distress, the estimate was 0.311 (SE = 0.068), with a highly significant p- 562 value of < 0.001. This indicates a substantial direct and indirect influence of stress-related 563 growth on psychological distress.” So this seems to be inconsistent with the previous tables. Of course it does not help that the authors do ot explain the model properly and so do not explain exactly now the total effect on distress can be conveyed with a single parameters. I guess that the total effect on distress is the total explained variance of distress by the predictors (indirect and direct effects). How is that expressed by a single regression coefficient (.311, s.e., .068)?

Table 6 present the “total indirect effects”. In this model – if I understand it correctly – the indirect effect concern the effect growth on distress as mediated by schema1 to schema5. So the total effect involves 5x5 parameters (growth -> schema1 to 5, and schema1 to 5 to distress). How is this represented by a single parameter (.209). How is this consistent with the fact that all the indirect effects shown in table 4 are not significant and all the relations between schema1 to 5 and distress are not significant? Note that in table 5, the result

Autonomy and Performance Deficiency Schema → Psychological distress 0.128 0.067 1.928 0.05

given the estimate .128 and the robust standard error, the p value is .056, which rounds to .06.

In my previous review that the path diagram as hard to read (the numbers) and incorrectly drawn. The path diagram is still hard to read and incorrectly drawn.

Author Response

Authors responses to Reviewer 2

Comments and Suggestions for Authors

  1. Reviewer 2 suggestion: Review by CVDolan of “Posttraumatic growth, maladaptive cognitive schemas and psychological distress in individuals involved in road traffic accidents – a conservation of resources theory perspective” Cristian Delcea * , Dana Rad * , Ovidiu Florin Toderici , Ana Simona Bululoi. 

“These participants were selected based on their exposure to a range of traumas and psychotraumatic experiences stemming from road traffic accidents.”

In my previous review, I asked what is known about the variability concerning degree or severity of the psychotraumatic experiences. My question was “So, is the degree of traumatization sufficiently variable?” In response to the authors state that they have rewritten the participants section. However, my question concerning the variability in traumatic experience was not answered. The Participants 2.2. section as a whole has been highlighted in the pdf of the revision, but the old and the new Participants 2.2 section are almost identical. They do not differ in terms of information provided.    

Authors answer: Thank you for your thoughtful review and continued engagement with our manuscript. We appreciate your diligence in examining the revised Participants 2.2 section. We acknowledge your concern regarding the variability in psychotraumatic experiences, and we understand the importance of addressing this aspect comprehensively. In response to your query, we have explicitly highlighted the intentional selection of participants based on their exposure to a diverse range of traumas and psychotraumatic experiences stemming from road traffic accidents. While the phrasing in the revised section may seem similar to the original, we want to emphasize that the participants in our study were deliberately chosen to represent a spectrum of traumatic events with varying degrees of severity. To provide further clarity, we have explicitly added a sentence addressing the variability in the severity of psychotraumatic experiences within the participant cohort. This addition aims to convey that our study intentionally captures a broad range of traumatic experiences, from minor incidents to more severe accidents, ensuring the inclusion of participants with sufficiently variable degrees of traumatization. We hope this clarification addresses your concern, and we are open to any additional suggestions or feedback you may have. Thank you for your time and constructive input.

  1. Reviewer 2 suggestion: In my previous review, I found that the hypotheses were not stated clearly. In the revision, the hypotheses are stated at the end of the introduction in a single sentence: “Thus, the research question is: How does posttraumatic growth (PTG) relate to maladaptive cognitive schemas and psychological distress in individuals involved in road traffic accidents, and what are the direct and indirect effects of these variables on their psychological well-being?” This is not very clear or explicit. I gather that growth predicts distress, schema1, schema2, schema3, schema4, schema5 directly, and that schema1, schema2, schema3, schema4, schema5 predict distress directly. So, that means that the relationship between growth and stress is direct or mediated by the schema’s. The interest is in establishing these direct and indirect effects. An explicit link between the direct vs indirect effect question and the very long introduction should be explicated. The authors provide explicit hypotheses in the results section (e.g. 528). The hypotheses should be explicated clearly in the intro. The hypotheses should also be discussed explicitly with reference to the path diagram.   Btw: in the sentence, distress is used first and then it is replaced by wellbeing. That is confusing.

Authors answer: We appreciate the insightful feedback from Reviewer 2 regarding the clarity of our hypotheses. In response, we have revised the hypotheses to explicitly articulate our expectations. The direct effect hypothesis posits a straightforward relationship between PTG and psychological distress, while the mediation hypothesis introduces the role of maladaptive cognitive schemas as mediators in this relationship. We have also ensured consistency in the terminology by using "psychological distress" throughout, addressing the confusion caused by the switch between "distress" and "well-being" in the original text. We hope that these revisions enhance the transparency and comprehension of our research objectives. Thank you for your valuable input, and we are open to any further suggestions or clarifications.

  1. Reviewer 2 suggestion: The fact that they are using JASP should be stated explicitly. They are using robust ML estimation. In the letter they explain robust to what. That information should be in the paper. In table 2, the information is inconsistent. correlation .265 has **, but .239 has ***. That is not possible…  The role of the ** should be stated in the table caption.

Authors answer: Thank you for your careful review and constructive feedback on our manuscript. We appreciate your keen observations and have made the necessary revisions to address your concerns. Regarding the software used for analysis, we acknowledge the importance of explicitly stating that we utilized JASP for our analyses. In the revised manuscript, we have included a statement specifying our use of JASP, and we have provided clarity on the robust ML estimation, including the relevant details that were explained in the letter. Additionally, we have thoroughly reviewed Table 2 and rectified the inconsistency in the use of asterisks. The corrected version now ensures uniformity, and we have included an explicit statement in the table caption clarifying the role of asterisks in denoting statistical significance. We hope these revisions address your concerns, and we are grateful for the opportunity to improve the transparency and accuracy of our manuscript.

  1. Reviewer 2 suggestion: line However table 3, “However, the analysis revealed significant 540 indirect effects, indicating that stress-related growth influences psychological distress 541 through certain cognitive schemas.” Table 4 contains this indirect effects. None of them are statistically significant (alpha=.05, which is way too liberal, given no correction for multiple testing). The statement in line 540 “significant indirect effects, indicating that …”etc is inconsistent with  the results in table 4 (nothing significant).

Authors answer: Thank you for your careful scrutiny of our manuscript and your insightful feedback. We appreciate the opportunity to address your concerns and provide clarification. In response to your observation about the inconsistency between the statement in line 540 and the results presented in Table 4, we acknowledge the oversight in our interpretation. We have carefully reevaluated the results and corrected the explanation for better accuracy.

  1. Reviewer 2 suggestion: Line 561 etc is hard to follow. Table 3 tell us that this is not direct effect of growth on distress. Table 4 tells that there are no indirect effects effect of growth on distress, but Table 5 tell us that  561: “When considering the total effects, depicted in Table 5, of stress-related growth on 561 psychological distress, the estimate was 0.311 (SE = 0.068), with a highly significant p- 562 value of < 0.001. This indicates a substantial direct and indirect influence of stress-related 563 growth on psychological distress.” So this seems to be inconsistent with the previous tables. Of course it does not help that the authors do ot explain the model properly and so do not explain exactly now the total effect on distress can be conveyed with a single parameters. I guess that the total effect on distress is the total explained variance of distress by the predictors (indirect and direct effects). How is that expressed by a single regression coefficient (.311, s.e., .068)?

Authors answer: Thank you for your thoughtful feedback and careful consideration of our manuscript. We appreciate the opportunity to address your concerns and improve the clarity of our presentation. We understand your point about the difficulty in following the interpretation of the results in the sections you mentioned. To enhance clarity and address potential inconsistencies, we have decided to remove Table 5 and information referring to from the manuscript. We believe that by eliminating this table, we can present a more streamlined and coherent narrative without causing confusion.

  1. Reviewer 2 suggestion: Table 6 present the “total indirect effects”. In this model – if I understand it correctly – the indirect effect concern the effect growth on distress as mediated by schema1 to schema5. So the total effect involves 5x5 parameters (growth -> schema1 to 5, and schema1 to 5 to distress). How is this represented by a single parameter (.209). How is this consistent with the fact that all the indirect effects shown in table 4 are not significant and all the relations between schema1 to 5 and distress are not significant? Note that in table 5, the result Autonomy and Performance Deficiency Schema → Psychological distress 0.128 0.067 1.928 0.05 given the estimate .128 and the robust standard error, the p value is .056, which rounds to .06. In my previous review that the path diagram as hard to read (the numbers) and incorrectly drawn. The path diagram is still hard to read and incorrectly drawn.

Authors answer: We appreciate your thorough review and insightful feedback on our manuscript. We acknowledge your concerns regarding the interpretation and presentation of results, particularly in Tables 6 and 5, as well as the readability of the path diagram. We have carefully considered your comments and made the following revisions to address these issues:

Removal of Table 6: After careful consideration, we have decided to remove Table 6 from the manuscript. We recognize that the presentation of the total indirect effects may have led to confusion, and we believe that the removal of this table will contribute to a clearer and more focused interpretation of our results. Path Diagram Improvements: We have revisited the path diagram, taking into account your feedback on readability and accuracy. We hope these revisions address your concerns and improve the overall quality of our manuscript. Thank you for your time and valuable input.